# A Retrospective Review of Global Commercial Seaweed Production—Current Challenges, Biosecurity and Mitigation Measures and Prospects

**DOI:** 10.3390/ijerph19127087

**Published:** 2022-06-09

**Authors:** Rajeena Sugumaran, Birdie Scott Padam, Wilson Thau Lym Yong, Suryani Saallah, Kamruddin Ahmed, Nur Athirah Yusof

**Affiliations:** 1Biotechnology Research Institute, Universiti Malaysia Sabah, Kota Kinabalu 88400, Sabah, Malaysia; mz1812001t@student.ums.edu.my (R.S.); wilsonyg@ums.edu.my (W.T.L.Y.); suryani@ums.edu.my (S.S.); 2Seadling Sdn. Bhd., Kota Kinabalu Industrial Park, Kota Kinabalu 88460, Sabah, Malaysia; bsp@seadling.com; 3Borneo Medical and Health Research Centre, Universiti Malaysia Sabah, Kota Kinabalu 88400, Sabah, Malaysia; ahmed@ums.edu.my; 4Department of Pathology and Microbiology, Faculty of Medicine and Health Sciences, Universiti Malaysia Sabah, Kota Kinabalu 88400, Sabah, Malaysia

**Keywords:** macroalgae, seaweed disease, climate change, ice-ice disease, seaweed farming, biosecurity, seaweed probiotics

## Abstract

Commercial seaweed cultivation has undergone drastic changes to keep up with the increasing demand in terms of the quantity and quality of the algal biomass needed to meet the requirements of constant innovation in industrial applications. Diseases caused by both biotic and abiotic factors have been identified as contributing to the economic loss of precious biomass. Biosecurity risk will eventually affect seaweed production as a whole and could cripple the seaweed industry. The current review sheds light on the biosecurity measures that address issues in the seaweed industry pushing towards increasing the quantity and quality of algal biomass, research on algal diseases, and tackling existing challenges as well as discussions on future directions of seaweed research. The review is presented to provide a clear understanding of the latest biosecurity developments from several segments in the seaweed research, especially from upstream cultivation encompassing the farming stages from seeding, harvesting, drying, and packing, which may lead to better management of this precious natural resource, conserving ecological balance while thriving on the economic momentum that seaweed can potentially provide in the future. Recommended breeding strategies and seedling stock selection are discussed that aim to address the importance of sustainable seaweed farming and facilitate informed decision-making. Sustainable seaweed cultivation also holds the key to reducing our carbon footprint, thereby fighting the existential crisis of climate change plaguing our generation.

## 1. Introduction

Seaweed is one of the world’s important marine source commodities, with a remarkably high species diversification, and it is prized for its nutritional and health benefits. For decades, these valuable marine resources have been harvested and grown for food and food ingredients across the globe by many groups of people, including Europeans, Asians, and South Americans. In 2018, 50 countries are engaging actively in seaweed farming with farmed seaweeds represented 97.1% by volume of the total of 32.4 million tons of wild-collected and cultivated aquatic algae combined [1,2]. Seaweeds are also known as benthic marine microalgae. They are categorized into three major groups, red (Rhodophyta), brown (Phaeophyceae), and green (Chlorophyta) based on the pigment molecules in their chloroplasts. Although they are empirically divided into three groups based on their color, seaweeds have different characteristics such as in the structure of the chloroplasts, cell wall composition, and photosynthetic pigments as well as many others [2,3]. At present, about 10 species of seaweed are farmed extensively among more than 200 known commercial seaweeds, which includes red seaweeds (*Eucheuma* spp., *Kappaphycus alvarezii*, *Gracilaria* spp., *Porphyra* spp.); brown seaweeds (*Saccharina japonica*, *Undaria pinnatifida*, *Sargassum fusiforme*); and green seaweeds (*Enteromorpha clathrata*, *Monostroma nitidum*, *Cauleurpa* spp. [3].

Prior to 2012, the global production of brown seaweeds by weight was higher than the production of red seaweeds [1]. The current trend shows that the red seaweeds dominated the market, due to their production of industrially important food ingredient hydrocolloids (carrageenan and agar). They are farmed in the Philippines and Indonesia and spread throughout other countries, including Malaysia, Tanzania, Brazil, the Solomon Islands, Fiji, Kiribati, India, and Mexico [4]. *Kappaphycus* and *Eucheuma*, also known collectively as eucheumatoids, are commercially important red seaweeds that are produced in high volume for global demand [4,5]. China remains the largest seaweed producer of *Pyropia*/*Porphyra*, *Gracilaria*, and *Kappaphycus* [5]. Other commercially important seaweed species produced by China for food include the brown algae *Undaria pinnatifida* and *Saccharina japonica*, which are some of the earliest seaweeds that have been cultivated using modern farming systems [6]. Both brown seaweeds are economically important commodities to other East Asian countries such as Japan and Korea [7,8]. Seaweed production is still in its infancy in Latin America, but the United States, Canada, and parts of Europe rely heavily on seaweed harvesting for the food and food polymer industries [5,9]. Seaweed mariculture has increased tremendously in the past decade due to its soaring demand as food, nutritional and bioactive ingredients, and other bioindustrial uses [2]. It is currently accepted that seaweed’s fast-growing capability with relatively zero use of arable land and chemicals (fertilizers, pesticides, and hormones) and less manual labor compared with terrestrial crops could provide us with a sustainable source of biomaterial and biomass for the future [10,11].

A report by the World Meteorological Organization showed troubling insight into the rapid decline of our climate over the recent 5-year study period. Global temperatures from 2015–2019 were the highest recorded, with an increase of 0.2 °C from the previous 5 years. Apart from that, there was an increase in emissions of greenhouse gases, such as CO_2_ and enteric methane (CH_4_), by 18% and 21%, respectively, while ocean temperatures rose by 0.13 °C [12,13]. The mass cultivation of autotrophic seaweed could help in alleviating the impact of climate change. Seaweeds are believed to be efficient carbon sinks that can absorb large quantities of carbon dioxide (CO_2_) from the atmosphere and prevent ocean acidification [14]. We need to fully take advantage of the benefits of seaweed that present probable solutions to combat global food shortages and loss of agricultural land and to mitigate climate change impacts in line with the United Nation’s Sustainable Development Goals (SDGs). This review highlights the challenges and the ongoing efforts to improve seaweed cultivation and production, including technologies to improve seaweed quality and disease mitigation, particularly in seaweed-producing countries.

## 2. Seaweed Uses: A Commercial Perspective

Current existing technologies allow for developing diverse uses of seaweed aside from food and food ingredients. Hundreds of different components can be sourced from seaweeds with tremendous industrial applications (Figure 1). Apart from the 77.6% of seaweed produced that is directly consumed as food, 11.4% is used in the phycocolloid sector (i.e., agar, carrageenan, and alginates) as food stabilizers and texture modifiers [2]. Green seaweeds are known to contain a significant amount of starch, with commercial species such as *Ulva ohnoi* (sea lettuce) containing up to 21.44% per dry weight [15,16]. There is a large number of dietary fibers found in commercial seaweeds that can be used as functional ingredients [17]. Seaweed could be a sustainable protein source to complement animal proteins that are increasingly scarce because the high dependence on terrestrial animals has caused immense environmental impacts [18]. For example, red seaweed *Porphyra purpurea* (nori) and brown seaweed *Undaria pinnatifida* (wakame) can contain as much as 33.2% and 16.8% protein per dry weight, comparable with some of the protein-rich terrestrial plants [19]. The unique sensory quality of seaweed also makes it useful as beverage ingredients and condiments [20,21].

Micronutrients and bioactive compounds sourced from commercial seaweed are valuable for the pharmaceutical and cosmetic industries. Ingredients such as amino acids (serine, palythine, shinorine, usujirene), vitamins (C, E, and retinol), iodine, and secondary metabolites (eckol, dieckol, astaxanthin, β-carotene, fucoxanthin, zeaxanthin, violaxanthin) are used in skin whitening and moisturizers, body shaping, UV protectants, dyes, and fragrances [22]. Additionally, similar bioactive compounds are used as therapeutic agents to regulate diseases such as metabolic syndrome (*Met*S). Fucoxanthin and astaxanthin (natural xanthophylls found in brown algae) are being utilized as commercial drugs with antioxidant, cholesterol reducer, anti-obesity, and anticancer properties [23,24]. They are significantly better compared with their synthetic competitors in many ways, including the safety concerns of using synthetic drugs on humans [25]. One of the commercialized bioactive polysaccharide components found in brown seaweed is fucoidan, which has anti-inflammatory and anticancer properties [26]. Interestingly, fucoidans from *Saccharina latissima* and *Laminaria hyperborea* were identified as the most promising candidates as potential therapeutics for age-related macular degeneration (AMD) [27]. In addition, seaweed alginate and carrageenan are used to produce polysaccharide hydrogels in tissue regeneration and wound healing [10]. In addition, seaweeds are excellent sources of minerals such as sodium (Na), potassium (K), calcium (Ca), magnesium (Mg), and phosphorus (P) [28] that are readily incorporated into dietary supplements, with the advantage of having a mostly low Na/K ratio, an important aspect of hypertension management [29].

Seaweed supplementation in livestock, poultry, and aquaculture feed has been practiced for decades and has shown tremendous improvements in animal health and meat quality and in reducing methane emissions of cattles [30]. For instance, the addition of brown algae *Undaria pinnatifida* and *Ascophyllum nodosum* in the diet of pigs and sheep improves the animals’ intestinal health, while the incorporation of red algae *Eucheuma denticulatum* in fishmeal improves the meat quality of Japanese flounders by increasing the amount of omega-3 fatty acids in the muscles [31,32,33]. Seaweed extract is commercialized as plant biostimulants as well. Biostimulants rendered by *Kappaphycus alvarezii* sap provide nutrients and plant growth hormones that have shown promising results in increased tomato growth (stem and root) and fruit yield and decreased disease and insect predation [34,35]. The research on sap utilization in land-based crops like legumes, maize, and wheat all showed positive results [36,37], with the ability to significantly reduce the need for chemical fertilizers. The application of fermented brown seaweed *Ascophyllum nodosum* fertilizer in the soil is also known to improve the microbial community and increase the nitrogenous species (NO^3−^, N, NH^4+^) in the ground [38].

Other uses of seaweed include its application in the energy and agricultural industries. The conversion of seaweed into biofuel such as bio-alcohols (ethanol and butanol) and biodiesel is made possible through the breakdown and conversion of the phytochemical components and the use of microbial fermentation [39,40,41]. Biofuel yields vary significantly across different seaweed species, fermentative microorganisms, and the technology employed, which directly affects conversion efficiency and cost, slowing down the efforts to realize a more feasible commercial-scale production [41,42,43]. However, there is a significant market potential for green biodegradable packaging material that seaweed can provide. Composite biopolymers made from seaweeds such as film have comparable strength, elasticity, and moisture permeability with petroleum-based plastics [44,45,46]. Several start-ups have created innovative biodegradable packaging materials made from seaweed to replace single-use plastic such as Notpla, Evoware, Algeon Materials, Sway, and FlexSea [47,48,49,50,51]. Green seaweed *Ulva* sp. can be a feedstock to produce polylactic acid (a bioplastic component) with minimal impact on the environment [52]. The current processes may not yet be highly cost-effective, but the battle to reduce the environmental impacts caused by conventional plastics needs to start with a mindset change of both industry players and consumers.

## 3. Seaweed Aquaculture and Sustainable Development Goals

The Sustainable Development Goals (SDGs) were introduced during the 70th session of the United Nations General Assembly, paving plans toward uniting all countries in building partnership and implementing strategies that improve health and education, reduce inequality, and spur economic growth—all while tackling climate change and working to preserve our oceans and forests [53]. Seaweed aquaculture contributes to the economies of coastal communities by employing millions of people, with 96% of all aquaculture engagement located primarily in Asian countries [1]. Seaweed’s upstream and downstream activities benefited different industries and spurred innovation and economic growth [6,38,54,55]. Seaweed farming provides positive effects on ecosystem services including supporting services (biogeochemical cycling, primary production, food web dynamics, biodiversity, habitat, and resilience), regulating services (atmospheric, sediment retention, eutrophication, biological regulation), provisioning services (food, raw material resources, energy), and cultural services (recreation opportunities, aesthetic values, science and education, inspiration, cultural and natural heritage, inspiration) [56].

The term “phyconomy” has recently been introduced to embrace large-scale, sustainable seaweed farming for economic benefit in coastal waters [57]. The positive prospects of phyconomy include human resource capacity enhancement, livelihood diversity, better ecosystem management, and operations sustainability such as resilience to climate change as well as global food security. Studies in Indonesia [58,59] and India [35] showed that coastal and island communities have benefited from their national seaweed farming policies to increase seaweed production and domestic processing. Seaweed cultivation provided families with a reliable supplementary source of income, improving their living standards and their children’s access to better education as well as empowering women as income earners. Reductions in fishing activities as a result of recruiting fishermen to seaweed farming in the Philippines could improve coastal fish populations [60], and sustainable seaweed harvesting in reef beds in Ireland [61] is among the ways of maintaining a balanced ecosystem. A similar pattern is also happening in Maine, where lobster fishermen are being recruited as kelp farmers as another option to make an income and also as a way to adapt to climate change. These are some examples of successful models executed with sound policies. Seaweed farming generally has a low negative impact on the marine environment when compared with other types of aquaculture such as fish and shrimp farming. However, commercial farming of nonindigenous stock of seaweeds may affect, directly or indirectly, important habitats such as seagrass beds and coral reefs [62]. Moreover, the constant introduction of foreign species in the open water systems might increase the risk of genetic loss of wild seaweed populations and become invasive, resulting in a detrimental threat to marine biodiversity [63]. Hence, there is a knowledge gap to fill by investigating the impacts of farming nonindigenous seaweed such as the environmental effects that might occur, whether there is any possibility of competing with other seaweeds for nutrients, and whether there will be any negative effects on fish assemblages [63]. The knowledge on how to manage farming nonindigenous seaweed to mitigate negative impacts is still limited. Hurdles in the form of lack of seaweed-related policy, lack of technological innovation and knowledge transfer, difficult access to financial means, and cultural and social issues exist and require immediate attention in some countries, which could hinder the realization of phyconomic success [9,57,64].

Climate change driven by human activities like greenhouse gas emission is shifting global weather patterns. There has been an increase in land and ocean surface temperatures and sea level and frequent occurrences of extreme weather [65]. Greenhouse gases were typically released into the atmosphere through the burning of fossil fuels, livestock farming, and forest burning. It was reported that farmed and wild seaweed can take up approximately 1521 TgC year^−1^ over an area of 3.5 million km^2^, contributing to the oceanic carbon sinks [66]. Seaweed is a natural biological CO_2_ absorber, with evidence suggesting that it could be a vital component in designing negative CO_2_ emission technologies keeping them in geological reservoirs [67,68,69]. Furthermore, researchers have suggested using seaweed in livestock feed to combat the issue of CH_4_ production that is mostly originating from agriculture farms [70,71]. Diet modification incorporating seaweed has been shown to affect enteric methane production and emission in ruminants whereby the generation of methane is reduced but with highly variable results depending on the composition and the type of seaweed used [30]. These studies further solidify the potential capacity of seaweeds as a vital component in reaching global SDGs (Figure 1).

## 4. The Emergence of Seaweed Diseases and Prevalent Threats to Seaweed Farming

The booming seaweed culture industry, particularly since the early 2000s, has resulted in an increased prevalence of diseases and aquaculture pests. The main commercial seaweed-producing countries are also countries that are heavily affected by seaweed disease outbreaks [3]. These outbreaks have caused a significant decline in hydrocolloid production, and the seaweed farming industry could collapse if left without any effective treatment or mitigation strategies [72,73]. Biosecurity measures refer to the controlling of diseases from the prevention of the accidental introduction of parasites and nonnative pests. It is important to implement effective biosecurity measures for protecting public health among consumers, ensuring the sustainability of the industry, and preserving the environment. Indeed, the biosecurity concept is not a new idea, although the majority of seaweed-producing countries pay little attention to it. It is only recently that studies are being directed towards understanding seaweed diseases and pathogens, their modes of infection, and their possible treatment methods. Generally, seaweed diseases are triggered by the presence of causative agents at an infected site, which activates the defense mechanism of the seaweeds to produce hydrogen peroxide. Prolonged infection changes the physiology of the seaweed, weakening its external structure and making it more vulnerable to further infection by other opportunistic pathogens [57].

The economic impact of disease outbreaks in commercial seaweed farms can be devastating, especially to farmers in developing countries. For example, a study in 2014 demonstrated that the disease outbreak in China resulted in an estimated loss of 25–30% of harvested brown algae, *S. japonica* (kombu), [73,74]. In Korea, *Pyropia* sp. (nori) farms suffered a loss of up to 20% while in the Philippines, a 15% decrease in *K. alvarezii* yield contributed to a financial loss of nearly $300 million between 2011 and 2013 [72,75]. These data clearly show how deeply the economy has been affected and will have a profound impact on the livelihood of farmers who are dependent on income generated from the sale of high-quality seaweed. Figure 2 illustrates the main identified threats affecting seaweed cultivation on open farms and the mitigation strategies undertaken.

### 4.1. Bacteria-Induced Diseases

There have been reviews on historical papers about the discovery of ice-ice disease and the probable causative agents involved [76]. Gavino Trono Jr. was the first to document ice-ice disease outbreak in the Philippines in 1974 [77]. In 1981, Flordeliz Uyenco and the team further discussed the biotic and abiotic conditions that contributed to the spread of the disease [78]. Seaweed infected with ice-ice disease can be easily detected by the visual observation of thalli that form white spots or bleaching (Figure 3). The name “ice-ice” was given as it looks like frozen thalli branches. Studies have generally narrowed down the disease-causing pathogens to *Vibrio* sp., *Alteromonas* sp., *Flavobacterium* sp., and *Cytophaga* sp. [76,78,79,80,81,82].

A pathogenicity test was adopted to identify the bacterial species associated with the disease, and three potential pathogenic microbes were isolated from infected *K. alvarezii* on Karimunjawa Island, Indonesia, i.e., *Alteromonas macleodii* (highest pathogenicity), *Pseudoalteromonas issachenkonii* (moderate pathogenicity), and *Aurantimonas coralicida* (lowest pathogenicity) [81]. Similar microbial results were recorded for other brown algae species in the northern Pacific Ocean. *Alteromonas* sp. was the causative agent of induced lesions in brown kelp, *Saccharina religiosa* (formerly *Laminaria religiosa*), in Japan [83], while the marine bacterium *Pseudoalteromonas issachenkonii* caused polysaccharide degradation in brown algae, *Fucus distichus* subsp. *evanescens* (*Fucus evanescens*), in the Kurile Islands [84]. The identified pathogenic bacteria could synthesize enzymes to degrade algal compounds and possessed flagella for successful seaweed colonization in water [78,85]. Surprisingly, there were other bacterial strains isolated from infected *K. alvarezii* belonging to *Bacillus* spp., *Pseudomonas* spp., *Rhodococcus* spp., and *Pseudoalteromonas* spp. that had little or no pathogenic effect [81]. There is a possibility that they might have formed a symbiotic or mutualistic relationship with the seaweed host.

Another example of a seaweed disease is the red-rot disease affecting Japanese and Korean *Pyropia* spp. farms caused by fungal oomycetes, *Pythium porphyrae*, and *Olpidiopsis* spp. Upon infection, the seaweed blades exhibit severe discoloration, appearing initially as red dots to light green and with perforation of varying sizes [73,86]. Although there have been extensive studies on the host-pathogen interaction, there is still little understanding of how to manage or prevent the spread of the disease. Adding to the complexity, two other causative agents, *Pythium chondricola* [87] and a plant fungus, *Alternaria* sp. [88], were identified as possible causes of red-rot disease. While researching the rotten thallus disease affecting the red algae *Gracilariopsis heteroclada* in the Philippines, various causative and noncausative agents were further discovered. *Vibrio parahaemolyticus* and *Vibrio alginolyticus* were detected in both healthy and diseased branches [89], suggesting that diseases are not only caused by a specific pathogenic species but may instead be a combination of multiple pathogenic bacterial interactions on a host coupled with other conducive environmental factors. Hence, studies related to seaweed diseases should comprise all groups of organisms that are directly or indirectly associated with the macroalgae to develop effective prevention and mitigation measures for disease outbreaks.

### 4.2. Epiphytic Attachment

Based on samples collected across Asian countries, the common epiphytes attached to seaweed surfaces (host) are identified as filamentous red algae (EFA) from the genus *Polysiphonia* and *Melanothamnus* (formerly *Neosiphonia*) [57,80,90]. Studies have shown that epiphytic algae use rhizoids to penetrate the cortex and medullary tissue of the host. Morphological observations during the initial stage of infection are the appearance of black spots followed by the development of red hair-like filaments. Seaweed tissue cells then disintegrate, forming cavities or pores that make them more vulnerable not only to secondary infection but also to herbivorous marine grazers [77]. The now-weakened thallus will break off from the main culture lines and float away [80,91]. These infections not only can be potential precursors to diseases, they can also adversely affect the photosynthetic activity and nutrient uptake in seaweed [92].

EFA outbreaks are prevalent in most Southeast Asian countries as well as in China and in Madagascar and Zanzibar in Africa [1,77,92]. In Sabah, Malaysia, two different EFA species were found, *Melanothamnus apiculatus* (formerly *Neosiphonia apiculata*) (Semporna) and *Melanothamnus savatieri* (formerly *Neosiphonia savatieri*) (Kudat) (Rhodophyta), with high infection rates during the inter-monsoon seasons (March-June and September–November). The same seasonal outbreaks were happening in the Philippines as well, indicating an association with fluctuations in seawater salinity and temperature [93,94]. An example of an EFA attachment on *K. alvarezii* is shown in Figure 4. There are at least two theories of how EFA outbreaks occur in cultivation farms: first, through contact between cultured *Kappaphycus* seaweed and free-floating EFA-infected *Sargassum* sp. and second, through imported cultivation stocks that were infected [95,96,97]. Other opportunistic biofoulers, such as the marine red algae *Acanthophora* spp. and *Laurencia dendroidea* (formerly *Laurencia majuscula*), have taken the opportunity to attach to infected seaweeds that are rich in halogenated secondary metabolites produced as a defense mechanism against bacterial infection in the event of an outbreak [98].

### 4.3. Herbivory Grazing

Herbivorous grazing is another factor impacting seaweed yields, though not as significant as disease or epiphytic infections. Historically, uncontrolled grazing by fishlings and juveniles like *Acanthurus dussumieri* (surgeonfish) have resulted in a loss of 50–80% of *Eucheuma* at a depth of 0.5 to 2 m in seaweed farms [99]. A review was done to classify these herbivores according to their feeding habits. For instance, “tip nippers” was given to fish that ate thalli tips, and “pigment pickers” were mainly juveniles consuming pigmented cell layers, while “thalli planers” were primarily sea urchins and green turtles that eat whole seaweed propagules [90]. Another approach was identifying the grazers according to the bite mark morphology; small holes on the thallus could have been caused by invertebrates, whereas irregular, jagged edges on the thallus are bite marks from sea urchins [100]. Herbivory grazing in eucheumatoid farms in China was reported with *Siganus fuscescens* (rabbitfish) being the dominant grazer, exhibiting a preference for red and green *K. alvarezii* and *E. denticulatum* due to their delicate morphological structures [92]. Marine amphipods were also found on cultivated commercial seaweed species in Japan. The isopod *Cymodocea japonica* grazed on the sporophyte of *Undaria pinnatifida* [101] and *Sargassum* spp. [102]. Listed are just a few of the many reported examples of biological issues faced by seaweed farmers who culture seaweed in open waters. 

### 4.4. Virus Infection

As mariculture practices worldwide intensify for commercialization, viruses can be introduced easily into the marine environment. At present, numerous accounts of diseases are emerging among seaweed farms, making it difficult to pinpoint the exact cause. Virus infections from the family Phycodnaviridae have been reported in *Ectocarpus* sp., a filamentous brown alga, and in kelp species like *Ecklonia* spp., *Laminaria* sp., and *Saccharina* sp. [103,104], while virus-like particles (VLPs) were discovered in tumor cells of the filamentous red algae *Bostrychia* sp. and in the red macroalgae *Delisea pulchra* [105,106]. Free-living gametes and spores of seaweed are highly susceptible to the viral DNA and RNA infections that are present in the surrounding waters. These viral loads can integrate themselves into the seaweed genome and remain highly active in the seaweed’s reproductive cells. This allows them to be passed down to the next generation, altering the genetic makeup and biogeochemical processes of future macroalgae [103,107,108]. Interestingly, when studying the green-spot disease (GSD) in Korean *Pyropia* spp. farms, researchers realized that primary infection began with lesions forming on the seaweed blades, followed by secondary infection with biofilm-producing bacteria [106]. This contradicted a study on Japanese *Pyropia* spp. farms that claimed that gram-negative bacteria were the leading cause of the disease [109]. Based on these two findings alone, we know that research should be focused on painting a big picture of the complex interactions between seaweed holobiont-like viruses and bacteria and their direct and indirect impacts on the hosts. It is also important to carry out a selective breeding program to develop certified virus-free seaweed strains that are pathogen resistant with the aim of developing fast-growing, disease-resistant seaweeds that minimize the risk of disease and maximize profits in cultivation.

### 4.5. Abiotic Factors

Disease studies on commercially important eucheumatoid seaweed have revealed a high density of bacteria present in infected branches. Seaweed branch tips inoculated with pathogenic bacteria whitened faster under prolonged low salinity exposure compared with branch tips without pathogens and applied environmental stressors [78]. This revealed the impact of environmental triggers on bacterial pathogenicity. Seagrass wasting disease, which is common in *Zostera marina* populations, is associated with the effects of environmental stressors including changes in sea surface salinity, temperatures, and ocean acidification [110]. The increased occurrence and severity of wasting disease outbreaks increase with salinity change, ocean warming, and light limitation [111,112]. A consequence of salinity and sea surface temperatures (SST) changes resulted in the release of high amounts of hydrogen peroxide (H_2_O_2_) by *K. alvarezii*, which may have increased its vulnerability to disease [57]. An increase in EFA infection by *Neosiphonia* sp. in *Kappaphycus* occurred when SST was at the lowest (28.9 °C) and salinity ranged between 31.7–32.1 ppt. These parameters coincided with the rainy season between July and September in Malaysia and the Philippines [93,94]. Ice-ice disease was reported in *Kappaphycus* farms in the Gulf of Mannar, India, during elevated temperatures (above 33.7 °C), high light intensity, and low water movement from March to April [82]. Herbivory grazing by marine isopods like *Ampithoe longimana* on *Sargassum filipendul* [113], *Peraphithoe parmerong* on *Sargassum linearifolium* [114], and *Cymodocea japonica* on *Undaria pinnatifida* (Phaeophyceae) has appeared to intensify with a 3 °C rise in water temperature [101]. All the mentioned cases indicate that all types of infections, whether bacterial, epiphytic attachment, or grazing including physical and chemical fluctuations in water properties, are contributing to disease outbreaks in commercial seaweeds.

Another abiotic factor to consider is nutrient abundance in the water column. It was initially theorized that nutrient abundance could trigger an increase in grazing behavior [115], although there was only a minimal effect of changes in carbon-nitrogen and carbon-phosphorus ratio, as a result of ocean acidification, on the overall microbiome of the brown seaweed *S. muticum* [116]. What could potentially take place is that a nutrient concentration elevation may alter the chemical properties of seaweed (carbon and sugar content), thus affecting its physical structure [117]. However, this statement requires further scientific validation. The ever-increasing threat of climate change will have a cascading effect on water pH and temperature, ocean stratification, and nutrient distribution throughout the water column. Extreme weather events like the increasing frequency and intensity of typhoons and prolonged monsoonal or drought seasons in seaweed-producing countries will take a toll on seaweed farm operations and the seaweed value chain [118,119].

## 5. Current Production and Farming Practices of Commercial Seaweeds

The commercial production of seaweed species varies heavily depending on the intended purpose, types of seaweed, desired traits, life cycle, cost efficiency, and geographical region [120,121]. The high demand for seaweeds in the past several decades has transformed its production, moving towards farming and cultivation nearshore, offshore, and inland. Asian countries such as China, the Philippines, Indonesia, Japan, and Korea are some of the leading counties where seaweeds are successfully produced through mass cultivation and are commercially exported worldwide. On the other hand, commercial harvesting of noncultivated seaweeds is still practiced in European countries, including Norway and Ireland. The cultivation of seaweeds in Europe, though, may not be feasible due to its labor-intensive methods, which makes it unprofitable [122]. The EU policies plan to utilize algae as third-generation biofuels are at an early stage of development and could only benefit the SDGs until technological limitations are addressed related to creating sustainable algae cultivation [123]. A comparable analysis in Chile initially reports nonfeasibility due to the high input cost, but the potential of its economic viability can be realized through the improvement of key factors such as market value, productivity, and the farming model [124,125]. Traditionally, seaweed cultivation involves the selection of desirable traits, especially higher growth rate, ease of manipulation, and high commercial value [126]. A higher growth rate will increase biomass yields during harvesting, and the high dollar value per unit of wet material would serve as an incentive for farmers to grow more. Nevertheless, for the farming systems to be sustainable at larger production scales, improvements are necessary, such as implementing effective communication among farmers in the biosecurity measurement aspects, developing systems capable of multiple partial harvests, and optimizing stacking and stocking density [122,127].

### Single Crop Farming and Harvesting Systems

Table 1 summarizes the production and single-crop farming systems of some commercial seaweeds. Large-scale commercial production of seaweed is mainly done in the open ocean or near the coastline. Commercial red seaweed such as *Pyropia* sp., *Gracilaria* sp., and eucheumatoid have been successfully cultivated using a long line or variants of the string-line method for several decades. The labor-intensive cultivation is mainly vegetative, usually requiring the manual attachment of seedlings onto a long rope or net, and is mostly practiced in Southeast Asian countries. Further improvements to the cultivation method for red seaweed include the implementation of net systems and cultivation in a floating cage or bottom-netted raft to prevent fish grazing [128]. Single-crop tank cultivation of commercial red seaweeds is possible, but due to the high cost, it is primarily for seedling production. The biochemical compositions of the ocean-grown biomass of *K. alvarezii* and *K. striatus* were not significantly different from that of the tank-cultivated biomass [129]. Regardless, the noncommercial cultivation of these seaweeds continues to occur among the coastal communities for their local consumption on a small scale.

Green seaweeds generally require substrates to attach themselves, so the method of cultivation differs slightly from the red seaweeds. The green seaweeds of the genus *Caulerpa* (sea grapes), for example, need sand or loamy substratum to attach themselves to using rhizoids and elongates, and they propagate by their stolon extension [134]. The small grape-like structure makes them known as “green caviar”, and the two most cultivated species are *Caulerpa lentillifera* and *Caulerpa racemosa*, which are prominent in countries such as China, the Philippines, Taiwan, Japan, Thailand, and Vietnam [134,145,146]. Tank aquaculture of these species requires the use of sandy or loamy bottom as a simulation of the alga’s natural habitat [132]. *Ulva* sp. is another commercialized green seaweed that has been cultivated using nets, offshore cages, and inland cultivation in tanks. Several *Ulva* species are consumed as vegetables, and its high biomass productivity makes it viable for large-scale cultivation [147]. The maximum daily growth rate of *Ulva* sp. was reported at 19.2% using offshore cages with constant tumbling and mixing of biomass with air and water exchange [136].

Brown seaweeds are commonly grouped as kelp. Some commercial species, such as *Laminaria* sp., *Saccharina* sp., *Sargassum* sp., and *Undaria pinnatifida*, have been successfully cultivated at a large scale using mostly the longline or rope method and variations. The production and farming strategies of kelp rely heavily on the kelp life cycle. Seaweed zoospores and gametophytes are used to seed onto substrates (lines, nets, ribbon, twines), which will then be transferred into the ocean [142]. Tanks for brown seaweed cultivation usually serve as nurseries to grow sporophytes and are not feasible for adult cultivation. The key difference in the mass propagation of brown seaweed is the source of explants from sporophytes compared with the vegetative fragmentation of red seaweed [5]. Both coastal and offshore farming along the Norwegian coast can produce 150–200 tons of kelp per hectare per year in an optimal growth period [148]. In 2018, the production of brown seaweeds accounted for 46.1% of the global seaweed aquaculture [1]. Wild kelp harvesting, which supplies *Laminaria* sp. and *A. nodosum*, is still commonly practiced in Norway and Ireland, respectively. *Laminaria hyperborea* harvesting in Norway on a commercial scale is typically achieved using trawlers in a sustainable revolving model with a steady annual harvest between 130,000 and 180,000 tons of wet weight per year [141]. *A. nodosum* accounts for around 95% of the total seaweed landing in Ireland, and it has been harvested mainly through manual cutting, rake harvesting, and mechanical harvesting in recent years [149].

## 6. Existing and Future Mitigation Strategies

### 6.1. Integrated Multi-Trophic Aquaculture (IMTA) Systems and Bioremediation Strategies

Integrated multi-trophic aquaculture (IMTA) systems are the next logical and innovative step to boost the production of seaweed. The system serves as a potential nutrient biological mitigation and bioremediation strategy in managing coastal farming areas under stress from drastic ecosystem changes due to the increase of aquaculture activities. The system can be applied in both open water and land-based cultivation. The concept of IMTA is the co-cultivation of fed species (finfish or shrimp) together with one or more species at a lower trophic level that acts as suspension-feeder (bivalves) and extractive species and carbon cycles (macroalgae). Each of the individual components in the IMTA system has to be of economic value for the system to be sustainable [150]. Seaweeds are the perfect combination in this system as they are known to be nutrient absorbers, especially nitrogen species [151]. The release of high-nutrient effluent from aquaculture farms without any form of wastewater treatment into the environment, especially into coastal waters, has been known to drastically change water quality, causing disruptions in the natural trophic state and potentially the spread of viral and bacterial diseases that can contaminate farms [152,153].

Several seaweed species, for example *Gracilaria*, that have been cultivated as part of the IMTA systems showed promising results in terms of mass yield, growth rate, and some noticeable changes in their biochemical properties when co-cultured with a vertebrate and an invertebrate species (Table 2), illustrated in Figure 5b. However, it is important to note that stocking density and seasonal variation affect the growth performance and chemical composition of the seaweed [154]. *Solieria filiformis* co-cultivation with sea cucumber and red-drum fish in tanks recorded the highest growth rate, 16.7%, from fish integration at day 70, with the highest protein content observed at day 30 being 20.1% with sea cucumber and red-drum fish combination [54]. High protein content in the culture tanks could turn the seaweed into a higher-value raw material in both the food and feed industries. The land-based co-cultivation of *K. alvarezii* with *Litopenaeus vannamei* (white leg shrimp) in a biofloc system showed not only a significant increase in growth rate but also an increase in bioactive components such as phenolics, flavonoids, and carotenoids [155]. The concept of a biofloc system, where water quality in an enclosed environment is controlled and managed by adding an external carbon source or microbes, is illustrated in Figure 5a.

IMTA cultivation in coastal areas showed promising results for both red and brown seaweeds. The longline cultivation of *K. alvarezii* and *Gracilariopsis lemaneiformis* (*Gracilaria lemaneiformis*) in a fish floating-net cage system exhibits growth rates at 3.33% and 5.82–9.84%, respectively [158,159]. These macroalgae not only serve as a biological mitigation solution but also provide additional income to farmers, as well as providing multiple ecosystem services such as oxygenation and carbon cycling [162,163,164]. The brown seaweed *Saccharina latissima* co-cultivated with *Salmo salar* (salmon) showed similar trends, with a 60% higher mass yield (1125 tons of fresh weight) compared with control and a higher protein content due to the estimated effluent absorption of approximately 11.8% of the 13.5 tons of dissolved nitrogen species [160]. Inorganic ammonia is the major dissolved nutrient from salmon farms, and the uptake of these species is beneficial for ecosystem balance [165]. A study on another brown seaweed *S. japonica* integrated into a poly species cultivation with *Mugil cephalus* (striped mullet) and *Patinopecten yessoensis* (yesso scallop) showed rather moderate daily growth rates between 0.03 and 1.9 mm/day with a blade length reaching 125 cm after two months of cultivation. The growth was highly affected by the water temperature, i.e., as the water temperature increased, the growth rate decreased, especially during the start of the summer month of May. IMTA systems prove to be a potential farming solution for the future to minimize the risk of monoculture farming, where all investments are placed in a single basket. IMTA promotes the diversification of aquaculture, which is desperately needed in the face of the changing climate and can serve as social and ecological security for aquaculture systems [166].

### 6.2. Disease Management Practices in Seaweed Farms

The presence of pests and diseases has plagued seaweed farmers for years. As a result, farmers have developed several cost-saving strategies to mitigate the issue. One of the commonly practiced methods used to combat diseases is washing seaweed blades in an acid solution for several minutes [72]. Others have resorted to handpicking any attached epiphytes as soon as possible from seaweed stock or harvesting seaweed with EFA attachment to avoid them from spreading [73,90,92]. Farm practices have also been adapted according to farmers’ experiences. For example, to avoid herbivorous grazers such as sea urchins and bottom-feeders, farmers either have the monolines submerged 50–100 cm from the surface or keep the lines away from coral reefs that harbor more predators [90,94]. *S. fuscescens* have no known natural enemies, so farmers employ gill nets or cages as a barrier to keep them away from the eucheumatoid farms [92]. *Porphyra* farmers freeze-store their nets at −20 °C to slow down the spread of red rot disease. The nets are carefully inspected for any signs of foreign attachments before being used for planting or storage [167]. Nevertheless, these are only temporary solutions to an age-long problem, and the introduction of more human-made materials into the water might contribute to pollution if they happen to drift off with the currents.

It became crucial for the aquaculture industry to have well-developed control and mitigation management plans in place in the event of a major disease outbreak. Many parties learnt a hard lesson for having a lack in biosecurity measures during the 1984 viral outbreak of infectious salmon anemia (ISA) in Norwegian farmed Atlantic salmon (*Salmo salar*) [168] and the 1992 white spot disease (WSD) outbreak in shrimp from Taiwan and China [169,170]. Subsequently, further emphasis was finally placed on improving local and regional legislation that is more country specific and on the introduction of biosecure hatcheries in developed countries to prevent the transboundary movement of introduced species [75,171]. Several biosecurity approaches and mitigations following the existing frameworks for fish and shellfish hatcheries should be identified and established for seaweed nurseries.

### 6.3. Genetic Manipulation and Strain Improvement

The idea of applying genetic manipulation for strain selection in seaweed culture was already being discussed in the 1990s, when domestic seaweed farming was pushed for commercialization. The goal was to produce seaweed strains that had a faster growth rate with higher carrageenan yields, particularly in eucheumatoids [90]. Widely used molecular techniques and advances in breeding tools to improve productivity and increase seaweed resistance to disease, predation, and epiphytism have been conferred in numerous research papers over the last two decades [57,172,173,174,175,176]. The adverse impact of global climate change on the commercialized seaweed industry has also driven genetic research to produce more thermo-tolerant strains [128,175].

The global demand for food is expected to increase by 70% by 2050 to accommodate 9.7 billion people, and under climate change scenarios, this requires projecting development opportunities of food production including aquaculture for future food security. Understandably, many considered GMOs a threat to the environment and human health. However, biases are ignored by GMO opponents in the pursuit of pushing a political agenda. The insistence on the possible negative effects of GMOs overshadows the very positive outcomes of useful plant traits induced by a human-driven method. Humans are facing a critical juncture where, on one hand, we are faced with unknown threats of GMOs to human health and the environment, while on the other hand, we have the chance, opportunity, and hope to change the way things are done conventionally. Therefore, instead of banning GMOs, regulations concerning the use of GMOs need adequate evaluation by breeders before marketing. The impacts of post-release GMOs should follow measures based on risk assessment and management. The continuous assessment of monitoring and detection methods is vital to enhancing the efficiency of food production and safeguarding environmental systems. Scientists should engage in the in-depth analysis of GMOs and other recombinant DNA agri-food methods, especially in the collection of ecological knowledge relevant to future releases. The use of GMOs is inevitable for meeting the increasing world demand and improving existing conditions in our environment. While the genetic transformation of macroalgae is further developed, careful strain selection, breeding, biosecurity, and certification standards would be more sustainable strategies for the future seaweed cultivation [177].

To adhere to the commercial restrictions from the US and EU on using genetically modified organisms (GMOs) in agriculture, scientists are researching natural and acceptable methods using advanced molecular techniques as an alternative measure to enhance seaweed production. As an example, researchers are actively experimenting on the bioconversion of compounds produced from microbial degradation to be used by seaweed to increase yield. Metagenomics was employed to identify potential genes and establish metabolic profiles of bacterioplankton involved in breaking down dissolved organic matter (DOM) and dissolved organic carbon (DOC) between reef communities [178,179]. Meanwhile, a genetically engineered alginate lyase enzyme secreted by tractable microorganisms on brown macroalgae was synthesized for alginate degradation in *Escherichia coli* to increase ethanol fermentation [180]. These experiments were only conducted in a laboratory setting. There are reports on the fermentation of seaweed monosaccharides by microbial activity, but it is not a simple task to identify microbes that can effectively metabolize them, and research is still ongoing in this area [181,182,183]. It is time that the EU regulatory legislation on GMO utilization in agriculture that has been in place since 2001 be reviewed again for the new Sustainable Development Goals by the UN to be achievable.

#### 6.3.1. Micropropagation and Hybridization

The 1990s brought together concerted efforts to develop background technologies for the micropropagation of a wide range of seaweeds, and several protocols for routine callus induction and regeneration are now available in the literature [57,184,185,186,187]. Clonal propagation of somatic embryos of seaweed was conducted in the hope of commercially cultivating them with the desired traits. The homozygous lines of *K. alvarezii* were maintained as filamentous calli with constant subculture on agar plates for more than two months before the regenerated micropropagules were out-planted at the farming site. These micropropagules exhibited higher growth rates compared with the wild seaweed cultures. The key to their success in seaweed tissue culture was to allow the plants to adapt to laboratory conditions before the experimentation [35,184]. Despite the advantages, many difficulties have been encountered in seaweed micropropagation, including the absence of optimized protocols for the acquisition of axenic cultures, callus induction, and the regeneration of whole plants using plant growth regulators [174,188]. Many researchers opposed using propagation with homogenous cultures as they produce culture lines that lack the genetic variation that is much needed to combat the challenges of climate change and disease outbreaks. The push for intensive breeding practices to meet market demand in Asia and Chile for *Gracilaria* and *Pyropia* strains unintentionally produced genetic uniformity, subjecting them to be more vulnerable to disease and weather conditions [128,189,190].

Hybridization is another technique used during cultivation to enhance macroalgae growth rates and biochemical characteristics for commercialization [76]. A new hybrid strain of *E. denticulatum* and *K. alvarezii* with significantly faster growth rates and novel carrageenan composition was developed using protoplast fusion and cell-cell fusion techniques ([172,175]. Fusion techniques allow for a wider variety of macroalgae to be crossed, which under normal circumstances is difficult to produce. The hybridization of kelp gametophyte *Saccharina longissima* (formerly *Laminaria longissima*) and *S. japonica*, a product of vegetative gametophyte crossing, in China has successfully produced new strains capable of withstanding higher irradiance, seawater temperatures, and tissue rot disease and that are tastier for consumption [126,190,191]. Similarly, the success of producing kelp hybrids between *Undaria peterseniana* (Kjellman) Okamura (male) and *U. pinnatifida* (female) in Korea enables a higher biomass yield and prolonged growing period including in the summer when the water is relatively warmer [7]. Several other hybridization techniques for seaweed propagation have been reviewed further [177]. It is important to note that there must be a comprehensive understanding of the genetic makeup and reproduction process of the seaweeds before conducting experiments on their gamete fusion. There is still a gap in the propagation of commercial eucheumatoids from spores to mass cultures, likely due to the lack of experts and optimized protocols in the field of seaweed reproduction [57].

#### 6.3.2. Molecular Identification for Strain Selection

The molecular identification of different seaweed species has gained momentum over the last 15 years as DNA sequencing technologies advanced rapidly and became more affordable. Researchers took the opportunity to build a seaweed database to create molecular markers and DNA barcodes to identify and characterize seaweed according to their phylogeny and phylogeographical locations [192,193,194,195]. This advancement was very much needed due to the complicated seaweed taxonomy and the lack of distinctive morphological features that made the identification of seaweeds, particularly from the important groups of *Eucheuma*, *Kappaphycus*, *Porphyra*, and *Gracilaria*, an extremely tedious task [192]. A genetic distinction was made for the first time between three red algae species, i.e., *K. alvarezii*, *K. striatus*, and *E. denticulatum*, from different parts of the world using mitochondrial cox2-3 and plastidial RuBisCo spacers [173] that were then used as DNA barcodes to differentiate seaweeds from both genera [196]. Examples of other molecular markers and DNA barcoding techniques used are simple sequence repeats (SSR) [197], random amplified polymorphic DNA (RAPD) [194,198,199], and inter-simple sequence repeats (ISSR) [200] to name a few. All these techniques have their advantages and limitations as more and more seaweed varieties and taxonomic groupings are discovered.

Because of the varying costs of DNA sequencing, more economically viable approaches have been developed, such as using designed primers from existing databases for PCR analysis for rapid species identification. RAPD was used to identify the genetic similarities and differences between green and brown varieties of *K. alvarezii* in Indonesia [194]. Interestingly enough, the authors discovered that there is a smaller difference in genetic variation between species with different-colored morphologies, whereas the genetic gap is widening between species from different sites of cultivation. This variation may be a result of phenotypic plasticity [173]. PCR techniques such as multiplex PCR using the internal transcribed spacer (ITS) sequence in the nuclear ribosomal DNA [195] and ISSR-PCR fingerprinting [200] have been used in the characterization of various strains of *Kappaphycus* and *Eucheuma* in Sabah, Malaysia. It is essential to gain insight into the taxonomic diversity of commercially cultivated strains to build a database for the potential cultivation of more disease-resistant strains or strains with desired traits. Studying seaweed-related diseases may be more manageable if species distribution is better understood.

#### 6.3.3. Induction of Stress Signaling to Increase Resistance

The oxidation of polyunsaturated fatty acids produces signaling molecules known as oxylipins. Derivatives of C18 and C20 oxylipin play a vital role in stress signaling for both plants and animals in abiotic (e.g., drought, heavy metal exposure, high temperature) and biotic (e.g., predatory, pathogen infection) stress conditions [201,202]. For example, the production of 13-hydroxyoctadecatrienoic acid, 15-hydroxyeicosa-pentaenoic acid, cyclopentenones (C18), 12-oxo-phytodienoic acid (C20), and volatile aldehydes was detected in brown kelp *Laminaria digitata* under stressed conditions [203,204,205]. The inducible lipoxygenase enzyme was also identified in kelp *Lessonia nigrescens* and the brown seaweed *Scytosiphon lomentaria* when subjected to copper stress [206]. Resistance in kelp *L. digitata* towards the endophytic pathogen *Laminariocolax tomentosoides* was induced through the production of arachidonic acid, linolenic acid, and methyl jasmonate (MeJA) [207]. Alternately, an unsuccessful attempt to induce C18 and C20 oxylipin production in the brown alga *Ectocarpus siliculosus* to enhance its resistance to the pathogen *Eurychasma dicksonii*. indicated that multiple defensive pathways may be responsible for activating stress signaling in the algae [202]. So far, the documented techniques for inducing stress-signaling mechanisms in land plants and brown seaweed are herbivory grazer-induced priming that produces unintended “stress imprints”, the transcriptional upregulation of defense genes, and the regulation of siRNA to targeted plant defense pathways [208,209].

### 6.4. Enhancement through Biostimulant and Biocontrol Strategies

#### 6.4.1. Seaweed-Derived Biostimulant

Biostimulants are complex mixtures of chemicals (e.g., humic substances, complex organic materials, protein hydrolysate, inorganic salts) and other natural products derived from biological process or extracted from biological materials (e.g., algae, microbes, fungi, plants, and animals) [210,211]. They were mainly used in the past to increase plant resistance to abiotic stress. As research progressed to understand the mode of action, the agricultural industry began to apply biostimulants (particularly with seaweed extract) to build resistance to biotic stressors [212]. Very little is known about the stimulatory effects of biostimulants as their mode of action may be synergistic with other existing compounds [213]. Studies have found that liquid seaweed fertilizers contain natural phytohormones such as cytokinins or gibberellins, vitamins, amino acids, and antibiotics [214,215,216], and others reported high amounts of auxin indole acetic acid (IAA), kinetin, zeatin, abscisic acid, ethylene, betaines, and polyamines [217,218,219].

The Ascophyllum Marine Plant Extract Powder (AMPEP), formally known as the Acadian Marine Plant Extract, is a biostimulant derived from *Ascophyllum nodosum* brown algae, commonly found on rocky intertidal shores in Atlantic Canada and northern Europe [212,220]. AMPEP is highly commercialized and used in the agricultural industry due to its efficacy in improving plant growth rates and enhancing resistance to abiotic and biotic stressors [94]. Traditionally, seaweed extracts have long been used as soil conditioners and foliar sprays for land-based agriculture around the world [221]. It is only now, as research continues to expand, that they have been classified as biostimulants with numerous benefits for both land and marine plants. Laboratory experiments showed that the extract was not only effective in promoting growth in land-based plants, i.e., *Arabidopsis thaliana*, but also improved micropropagation and reduced epiphytic attachment by *Ulva* sp. and *Cladophora* sp. in *K. alvarezii* [222,223,224]. Due to their simple application, by submerging vegetative cuttings of seaweed thalli in culture medium containing AMPEP, and their easily observable effects, supplementation with biostimulants is the most studied approach to seaweed micropropagation.

*K. alvarezii* achieved the highest growth rate with less EFA infection when dipped in seawater containing 0.1 g/L^−1^ of AMPEP for 30 min before cultivation at a depth of 50–100 cm [94]. Additionally, AMPEP-treated *K. alvarezii* and *K. striatus* exhibited an increase in free radical scavenging and transition metal chelating abilities [225]. This indicates great potential for more research on the ability of seaweed extracts to enhance disease resistance and increase the uptake of abiotic substances from seawater. AMPEP products must be readily available to local seaweed farmers and markets, for its usage to be fully integrated into commercial seaweed farms [212]. In the interim, developing countries such as India, Africa, and Indonesia have been researching biostimulants made from seaweed sap. In India, this new locally produced biostimulant containing nutrients and plant growth hormones similar to AMPEP is being used as a foliar spray on selected agricultural farms across the country. Seaweed sap harvested from fresh *K. alvarezii* and *Gracilaria* sp. has shown promising results in increasing biomass and yield in land-based crops like legumes, maize, tomatoes, wheat, and rice [34,35,36,37,226,227,228]. Leaf curl and insect predation were also successfully reduced to a controllable level via the application of biostimulants. From an environmental standpoint, seaweed extract has the potential to eliminate once and for all the need for chemical fertilizers, thus avoiding decades-long concerns about environmental pollution and the reduced life quality of farmers.

#### 6.4.2. Biocontrol in Seaweed-Bacteria Interaction

*Pseudoalteromonas* and *Vibro* are two genera of bacteria that are of high marine ecological significance due to their broad bioactivity range of antimicrobial, antifouling, and algal degradation properties. Some examples of identified species from seaweed are *P. aurantia*, *P. luteoviolacea*, *P. tunicate*, *P. tunicata*, and *Vibrio* spp. [76,229,230,231]. However, their ability to act as antifoulers is not as simple and direct as it seems. Although *Vibro* sp. and *Pseudoalteromonas* sp. extract and biofilm managed to reduce the attachment of *Hydroides elegans* larvae, further individual bacterial testing revealed no significant results [79]. This led to the conclusion that there may be a complex relationship between the host and its seaweed holobionts that prevent biofoulers from being attached. The use of bacteria for antifouling activities, especially from *Pseudoalteromonas* and *Vibrio* species, has been reviewed in detail [107].

Microorganisms that function as inhibitors of fouling activity also play a role in the prevention of seaweed disease. The possible dynamics between biofoulers producing antibacterial metabolites with the occurrence of ice-ice disease was investigated. A dominant filamentous biofouler, *Lyngbya majuscula* (Cyanobacteria), which appeared to thrive during periods of ice-ice infection, produced antibacterial metabolites composed of elatol, iso-obtusol, (Z)-10,15-dibromo-9-hydroxy-chamigra-1,3(15),7(14)-triene, and (E)-10-15-dibromo-9-hydroxy-chamigra-1,3(15),7(14)-triene. These halogenated compounds inhibited the growth of *Cytophaga–Flavobacterium*, *Vibrio* sp., and *Alteromonas* sp. isolated from diseased branches [98]. Similarly, *Eucheuma* seaweeds are known to produce volatile halocarbons (VHCs) when subjected to environmental stressors such as pH or light intensity variations that appear to alleviate epiphytic foulers [92,232]. Biostimulants are a common topic in literature, but most studies focused on their uses on land-based agriculture instead of on combating diseases among commercial seaweed.

Even though there is a negative connotation towards herbivory grazing, rabbitfish, *S. fuscescens*, have exhibited a preference for consuming EFAs such as *N. savatieri* and *Obelia* sp., over *K. alvarezii*. As a result, farmers were able to substantially control the outbreak of EFAs occurring annually from May to August and in October on the *K. alvarezii* farms [92]. This finding shows that controlled herbivory grazing can in turn be beneficial for seaweed farms facing devastating destruction from an epiphytic outbreak.

## 7. Prospects and Recommendations

Our foods in the future will inevitably come mostly from the ocean, with seaweed playing a vital role in maintaining a constant food supply to feed the ever-growing world population. The vast expansion of seaweed aquaculture is also the most promising answer to addressing climate change issues and buffering the negative impacts of human activities. Efforts to address critical challenges in the upstream seaweed industry from seeding to drying and packing are always improving the quality and productivity of seaweed biomass [233]. These include increasing the composition of major commercial components (e.g., nutrients, hydrocolloids, and bioactive components) in seaweeds using genetically superior varieties and advanced aquaculture technologies. High-quality seaweed raw material is crucial to meeting the demand of the downstream application for products of higher commercial value [55]. The future seaweed research at the upstream level is expected to continue along with these areas, including farming and cultivation technologies, the genetic improvement and diversity of seedling/sporophyte stock, epiphytes and diseases, and impacts on ecosystem and biodiversity. Offshore farming offers infinite space in the ocean and can resolve problems associated with limited and expensive areas for land-based cultivation [234]. Additionally, the effective implementation of biosecurity measures and the rapid detection of disease and pests are crucial for the sustainability of the seaweed industry. Several projects such as GlobalSeaweedSTAR, Phycomorph, Genialg, Seaweed for Europe, and Eklipse brought together companies and experts in areas of the seaweed niche such as cultivation, genetics, biosecurity practices, metabolomics, and disease detection by providing funding and support to boost the seaweed industry. For instance, GlobalSeaweedSTAR’s research focused on disease and pest detection, biosecurity practices and policy, algal genetic resources, and socioeconomic resilience in the seaweed industry. In addition to research, this program offered funding for developing countries and the UK, online resources, and a series of events to allow the exchange of knowledge between industry stakeholders and researchers. Continuous projects such as these should be offered especially to the developing countries to build capacity or strengthen the seaweed industry. It is necessary, however, to first identify seaweed varieties that can tolerate the harsh environments in the open ocean. The effects of seaweed farming on both the benthic and coastal ecosystems requires constant monitoring as well. Intensive seaweed farming along coastal areas will affect benthic biodiversity as it will attract grazers and invasive cultivars that could exterminate native cultivars [121,235]. IMTA systems may continue to be the future of sustainable aquaculture through incorporating the processing of waste products from one trophic level into valuable co-products at a lower trophic level. Other benefits and services (e.g., absorption of excess nitrogen species, carbon sink) may also be generated by integrating “extractive species” (i.e., seaweed) in addition to increased product profitability [150].

Research into the molecular fingerprinting of desired seaweed varieties will continuously build a comprehensive system of genetic and metabolite databases for seaweeds [236]. A full database system that is readily available is essential for the accurate identification of seaweed species and varieties. The micropropagation of *Kappaphycus* and *Eucheuma* species through optimized tissue culture, seedling enhancement, and mass production protocols may be the key to increasing production [237]. The immediate challenge, however, is to achieve the lowest possible cost of high-quality laboratory seedlings [176]. Studies have shown that biostimulants enhance the growth of commercial seaweed such as in *Eucheumatopsis isiformis* [238] and *Kappaphycus* spp. [239]. With the growing body of evidence related to the positive impacts of using biostimulatory extracts to support the cultivation of seaweed crops, biostimulants can be used in seaweed production and health. The use of biostimulants such as AMPEP to enhance growth and disease resistance will continue to be explored across the commercial seaweed species. Another key area that is potentially important for generating quality seaweed biomass and disease resistance is to control the seaweed microbiome using specific host-bacterial/epiphytic relationships. Although this research area has received moderate interest from the scientific community, it is worth being reassessed for its potential benefit and a thorough understanding of its underlying mechanism. Although the host-pathogen relationship is highly complicated and challenging to study, many reviews and articles over the years have highlighted the enormous potential of using this relationship to the benefit of the host, in this case, seaweed [240]. Most host-bacteria interactions are beneficial or mutualistic unless the balance is skewed to favor one over the other. The exchange of nutrients and bioactive compounds between the seaweed host and both epiphytic and endophytic bacteria had been shown to improve the growth, development, and reproduction of seaweed by increasing their resistance to pathogens, opportunistic grazers, and marine foulers [76,230]. Additionally, the polysaccharide compounds found in these macroalgae are not present in other land or marine plants or animals, making these resident bacteria reservoirs for novel bioactive metabolites or enzymes [241,242,243].

Listed are a few of the discoveries made in response to using microbial communities to trigger the production of bioactive metabolites in seaweed, similar to their application in controlling diseases among land plants. A protein exhibiting antibacterial activity believed to be an N-AHLs compound involved in quorum sensing (QS) was discovered when studying growth-promoting activity in red seaweed *Gracilaria edulis* [244]. Similarly, a protein from two growth-promoting bacteria, *B. megaterium* GT119 and *Lysinibacillus xylanilyticus* GT132, in *G. edulis* exhibited similar QS characteristics against *Xanthomonas oryzae* pv. *oryzae*, a plant pathogen [242]. It is reasonable to find QS compounds during bioactivity as they are a mode of communication triggered when competition for space and nutrients increase in dense colonies of bacteria. Another example of bacteria exhibiting probiotic potential is the involvement of *Wenyingzhuangia* spp. (marine bacteria) in the breakdown of sulphated organic compounds in *E. denticulatum* and *K. alvarezii*, which may play a role in the nutrient exchange and the tolerance to environmental stress ([76,245]. An abnormal thallus formation in the cultivated green alga *Ulva lactuca* was restored via the inoculation of bacterial strains from the same individual seaweed [107]. The red alga *Gracilaria dura* showed a significant increase in budding when associated with the epiphytic bacteria *B. pumilus* and *E. homiense*, possibly caused by protein conjugated with IAA and nitrogen fixation [246]. The bacteriolytic capabilities of *Algicola bacteriolytica* and the marine bacteria *Ruegeria* could play a role in the growth and defense of *E. denticulatum* and *K. alvarezii* [76]. About half of the isolated epiphytic bacteria from the brown alga *Saccharina latissima* exhibited antimicrobial activity against at least one Gram-positive and Gram-negative bacteria [247]. There was an indication that seaweeds rely on the presence of epiphytes or endophytes to produce bioactive compounds when further testing on the crude extract of *K. alvarezii* revealed no antimicrobial activity [248].

## 8. Conclusions

Studies over the last three decades have shown remarkable growth in the seaweed industry and increasing global demand for seaweed products. All efforts have been aimed at improving quality and production yield, but similar efforts must be made to understanding the causes and prevention methods of disease outbreaks. Innovation in farming and cultivation techniques will continue to develop, not only as a way to increase the biomass of seaweed but also as an ecosystem service that improves the conditions of coastal waters against the impacts of climate change. Indeed, advances in molecular diagnosis in the seaweed industry have fallen behind compared with other aquaculture species. That was the case up until these past few years. We are currently seeing more studies on the dynamics of host-pathogen relationships, on the biotic and the ever-changing abiotic stressors that affect pests and disease outbreaks. The use of seaweed biostimulants has tremendous potential to increase the production yield and quality of land crops, as it has already been a success in numerous studies. The frameworks for biosecurity and the mitigations of inland crops could be adapted for seaweed cultivation. Hence, future researchers need to investigate the potential of seaweed biostimulants to increase productivity and reduce disease outbreaks in seaweed cultivation.

Innovation in seaweed-microbial probiotics and the sustainable use of seaweed as health foods is still in the developmental stages, as research continues to identify components and marine microbiome in healthy and infected seaweed through metagenomics studies. It is crucial to develop the right breeding strategies and seedling stock selection to ensure the genetic diversity of seaweed on cultivated farms with complete adherence to local and regional biosecurity measures and international GMO regulations. Local and international organizations need to work together to align policies and come up with a comprehensive framework that is focused on providing education and financial and technological support to seaweed farmers from developing nations. Eventually, there are thousands if not more coastal communities whose livelihoods depend on the long-term success of seaweed cultivation. That along with the economic and environmental benefits should be a driver for seaweed-producing countries to direct their investments towards building a more sustainable seaweed industry.

## Figures and Tables

**Figure 1 ijerph-19-07087-f001:**
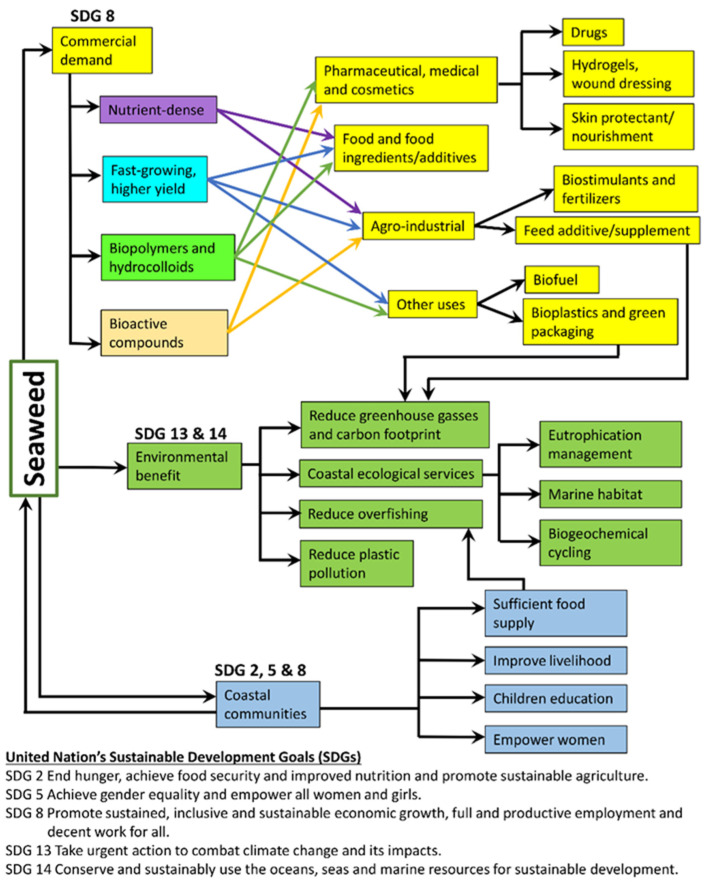
The intricate web of the relationships between commercially demanded seaweed attributes and their potential contributions to the United Nation’s Sustainable Development Goals (SDGs).

**Figure 2 ijerph-19-07087-f002:**
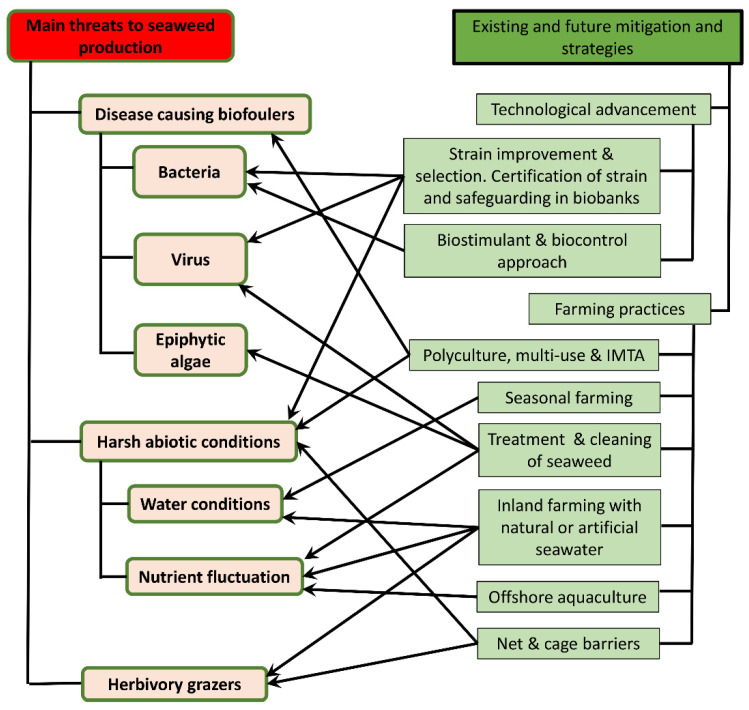
The main threats to seaweed production and their mitigation strategies.

**Figure 3 ijerph-19-07087-f003:**
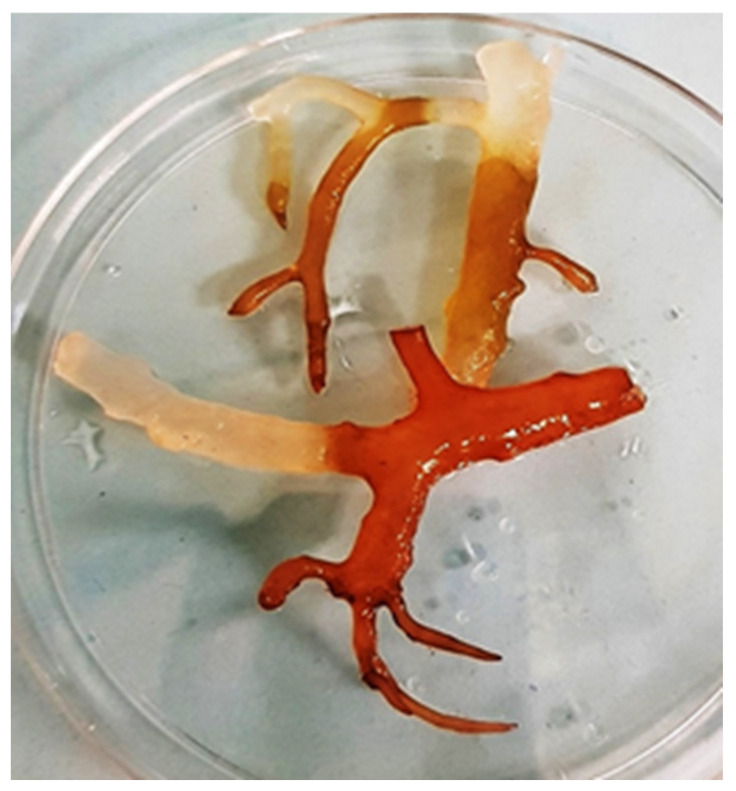
The presence of ice-ice disease or bleaching on thalli tips of *K. alvarezii* from Kota Belud, Sabah.

**Figure 4 ijerph-19-07087-f004:**
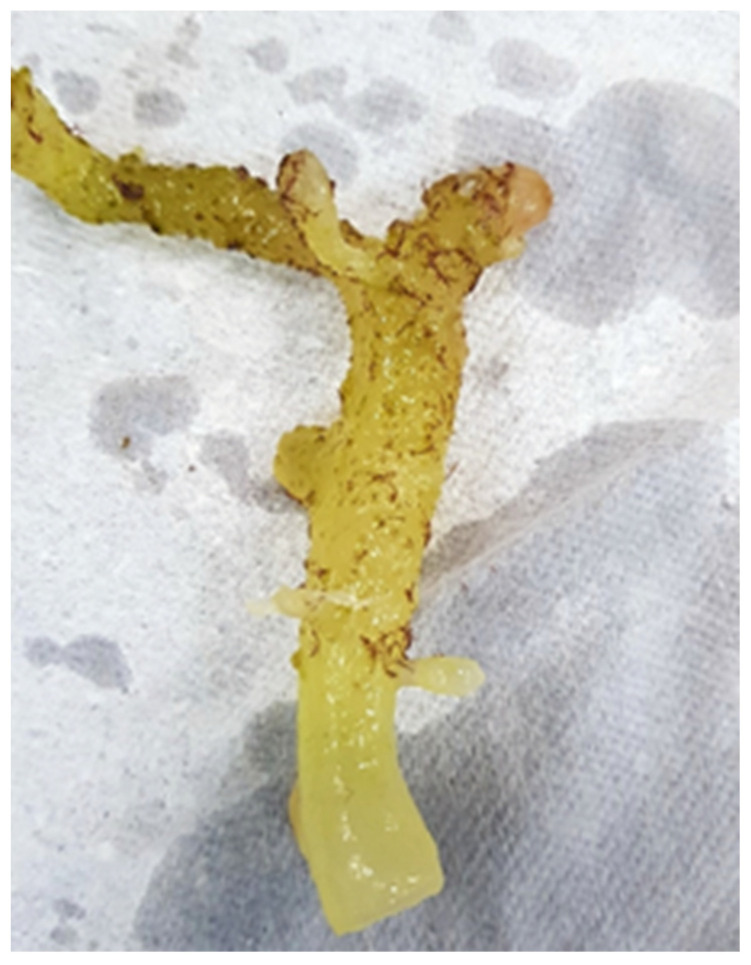
Signs of *Neosiphonia* infection on a *K. alvarezii* branch harvested from Kota Belud, Sabah.

**Figure 5 ijerph-19-07087-f005:**
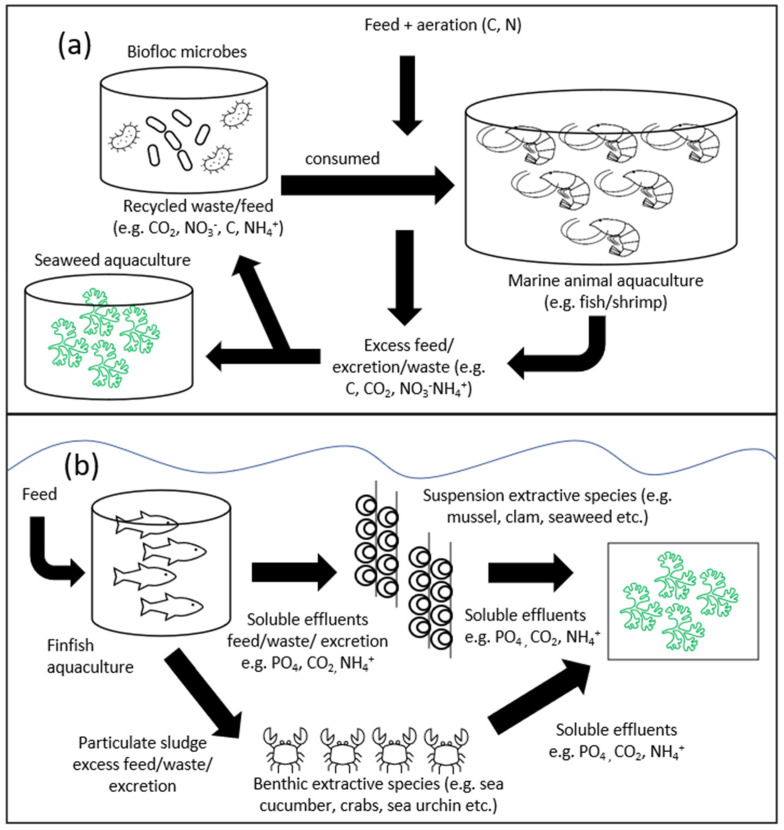
(**a**) A schematic diagram of a biofloc system implemented in an aquaculture enclosure. Co-cultivation of seaweed and biofloc microbes in the aerated system can consume and recycle excess feed and waste from the aquaculture species. (**b**) An IMTA system consisting of multiple species acting as extractors in which both can be adapted to improve seaweed growth rate and biomass quality.

**Table 1 ijerph-19-07087-t001:** Single-crop farming and harvesting system of some commercial seaweeds.

Seaweed Species (Seaweed per Color)	Production Method	References
*Porphyra* sp./*Pyropia* (red algae)	Commercial production offshore and nearshore using fixed poles, nets, semi-floating rafts, or floating rafts	[8,128,130]
*Kappaphycus* *alvarezii* and eucheumatoids (red algae)	Small-scale harvesting on reef beds Commercial offshore and nearshore production mainly using longline methods, bamboo rafts, and floating cage culture Inland tank cultivation	[131,132]
*Gracilaria/Gracilariopsis* (red algae)	Commercial scale nearshore using floating bamboo rafts with bottom netting, bottom cultivation, Inland production in pond and tank	[128,133]
*Caulerpa* *lentillifera,* *Caulerpa* *racemosa* (green algae)	Noncommercial harvesting on reef beds Commercial-scale farming using submerged rafts, longline methods Commercial-scale inland pond cultivation and tank cultivation with sandy loamy-substratum and water circulation	[132,134]
*Ulva* sp. (green algae)	Commercial offshore cultivation on nets, cages Inland cultivation in tanks	[135,136,137]
*Saccharina latissima*, *Saccharina japonica* (brown algae)	Offshore longline horizontal and vertical methods for commercial-scale production	[120,128,130,138]
*Sargassum fusiforme, Sargassum fulvellum* (brown algae)	Commercial-scale offshore farming using the longline method	[139]
*Undaria* *pinnatifida* (brown algae)	Commercial-scale offshore farming using longline, vertical hanging methods	[8,140]
*Laminaria* sp. (brown algae)	Commercial-scale harvesting of wild species using trawling tools Offshore cultivation using zoospores and gametophytes on nets and lines	[130,141,142]
*Ascophyllum nodosum* (brown algae)	Commercial-scale sustainable harvesting by mechanical and hand cutting on seashore beds	[143,144]

Nearshore: 500 m to 3 km fron the coast; Offshore: >3 km from the coast.

**Table 2 ijerph-19-07087-t002:** The IMTA systems and bioremediation strategies of offshore and inland cultivation of some commercial seaweeds.

Seaweed Species (Seaweed per Color)	Effluent Source	Effect of Growth and Quality of Seaweed	Reference
Inland cultivation			
*Kappaphycus alvarezii* (red algae)	*Litopenaeus vannamei* cultivation in a biofloc system	Significant growth rate of 1.70% day^−1^ compared with control; increase in total phenolics, flavonoids, and carotenoids; ice-ice disease observed in some samples	[155]
*Gracilaria verru-cosa* (red algae)	*Mytilus galloprovin-cialis (Mediterra-nean mussels) co-cultivation*	Maximum growth rate of 4.45% on day^-1^ during spring; reduction in water ammo-nium and phosphate concentration	[156]
*Gracilaria vermiculophylla* (red algae)	Fishpond effluent	Increased biomass and mycosporine-like amino acid (MAA) content during the summer months of April and May; MAAs also affected by stocking density	[154]
*Gracilariopsis longissima* (formerly *Gracilaria verrucosa*) (red algae)	*Mytilus galloprovincialis* (Mediterranean mussels) co-cultivation system	Maximum growth rate of 4.45% on day^−1^ during spring; reduction in water ammonium and phosphate concentration	[157]
*Gracilaria edulis*, *Gracilaria changii* (red algae)	Wastewater recirculation system from shrimp culture	Mean growth rates observed of 4.1–4.3% on day^−1^; removal of ammonium and nitrate at 71.0–72.5% and 56.8–58.8%	[151]
*Solieria filiformis* (red algae)	*Sciaenops ocellatus* (red drum fish) and sea cucumber integrated system	Significant increase in growth rates and protein content in the integrated system compared with control; highest growth rate recorded at 16.7% from fish integration on day 70; highest protein content recorded at 20.1% on day 30 in fish and sea cucumber integration	[54]
*Ulva lactuca*	*Rachycentron canadum* (cobia fish) and *Perna perna* (brown mussel)	Significant daily growth rate observed over time and inclusion of trophic levels; highest recorded at 4.75% (fish) and 6.32% (fish and mussel)	[157]
Ocean-based cultivation			
*Kappaphycus alvarezii* (red algae)	Fish (not specified) in floating net-cage system	Significant growth rate of 3.33% compared with control	[158]
*Gracilariopsis lemaneiformis (Gracilaria lemaneiformis)* (red algae)	*Pseudociena crocea* (yellow croaker) in floating cage system	Specific growth rates of 5.82–9.84%; total weight at 35 days was 5.3 times than initial weight	[159]
*Saccharina latissima* *(brown algae)*	Integrated farming with *Salmo salar* (salmon)	Increased macroalgae yield by 60% with potentially higher protein content compared with nonintegrated systems for a 25-hectare farm.	[160]
*Saccharina japonica (brown algae)*	Integrated farming with *Mugil cephalus* (striped mullet) and *Patinopecten yessoensis* (yesso scallop)	Daily growth rates ranged from 0.03–1.9 mm/day; highly affected by seasonal water temperature	[161]

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
