# Peer review of "A Retrospective Review of Global Commercial Seaweed Production—Current Challenges, Biosecurity and Mitigation Measures and Prospects"

_ijerph, 2022, doi:10.3390/ijerph19127087_

Round 1
Reviewer 1 Report
The authors introduced the corrections suggested by the reviewers, so I only detected a few errors, which should now be corrected, as I suggest below.
Corrections needed:
line 42 - red algae (Rhodophyta), brown algae (Phaeophyceae) and, green algae (Chlorophyta) (Note: The only taxonomic categories that should be written in italics are the genus and the species, and the categories below the species. The phyla and classes are written in normal letters)
line 312/313 - found, Melanothamnus apiculatus (formerly Neosiphonia apiculata) (Semporna) and Melanothamnus savatieri (formerly Neosiphonia savatieri) (Kudat) (Rhodophyta), (Note: The only taxonomic categories that should be written in italics are the genus and the species, and the categories below the species. The phyla and classes are written in normal letters)
Table 2 - Gracilariopsis longissima (formerly Gracilaria verrucosa)
Author Response
First, we would like to thank the reviewer for his/her tremendous efforts in improving our manuscript. We hope that this review paper will assist those who work in this line to use effective methods in developing sustainable seaweed farming. We appreciate the time and effort that the reviewer had dedicated to providing valuable feedback on our manuscript, and are grateful for his/her constructive comments and kind words. We sincerely appreciate all your valuable comments and suggestions. We have amended our manuscript accordingly.

Reviewer 2 Report
The authors have revised the manuscript accordingly and greatly improved the text.
Author Response
First, we would like to thank the reviewer for his/her tremendous efforts in improving our manuscript. We hope that this review paper will assist those who work in this line to use effective methods in developing sustainable seaweed farming. We appreciate the time and effort that the reviewer had dedicated to providing valuable feedback on our manuscript, and are grateful for his/her constructive comments and kind words. We sincerely appreciate all your valuable comments and suggestions. Thanks
This manuscript is a resubmission of an earlier submission. The following is a list of the peer review reports and author responses from that submission.
Round 1
Reviewer 1 Report
Please see the attached document

Reviewer 2 Report
Dear Authors,
This manuscript presents a comprehensive review of the subject, is well written and organized, comprising the abstract plus seven specific sections, besides the introduction, conclusion and references. The reference list is extensive and valuable and corresponds with the citations.
Concerning the selection key-words, it is not obvious the reason for including one specific genus (of red seaweed Kappaphycus) in detriment of the other mentioned in text.
In what concerns headings, avoid using full stops (eg. Section 3) and replace “n.0” by “n.” (Sections 4-9).
Please revise numbers for in text citations: eg. first in text citation of [69] appears (in line 272) after first in text citation of [70] (in line 225).
Please see below some minor comments.
- Introduction
Please include in this section the meaning of “eucheumatoid”/“eucheumatoids”, used in the subsequent sections (lines 302, 332, 390, 508, 587 and table 1).
Line 35 – Please rephrase. FAO report from 2020 mentions “In 2018, farmed seaweeds represented 97.1 percent by volume of the total of 32.4 million tonnes of wild-collected and cultivated aquatic algae combined”. So the value 32.4 million tons refers to seaweed production in 2018 (both farmed and wild) and this is a fact, not an estimate; besides farmed seaweed represented 97.1% of the volume of 32.4 million tons. Also, please include the year when this production was achieved (2018).
Line 46 – Please replace “Pyopia” by “Pyropia”
- Seaweed Uses: A Commercial Perspective
Lines 103-107 – Please rephrase the sentence beginning with “Fucoidan is also”, as it is very long and hard to understand.
Line 135 - Please replace “Ulva Sp.” by “ Ulva sp.”
- Seaweed aquaculture and sustainable development goals.
Line 177 - Please replace “TgC yr–1” by “TgC yr–1”
- The Emergence of Seaweed Diseases and Prevalent Threats to Seaweed Farming
Lines 211 & 248/ 323 & 325 - “Pyropia” : please state the species or use sp./ and use italics.
Line 235 - Please introduce a comma before Pseudoalteromonas
Lines 238 & 290 - Please use italics for “Kappaphycus” and Neosiphonia”.
Line 332 - Please replace “euchematoid” by “eucheumatoid”
Lines 347-350 – Please rephrase the sentence beginning with “All the mentioned cases”, as it is very long and hard to understand.
- Current Production and Farming Practices of Commercial Seaweeds
Line 390 - Please replace “euchematoid” by “eucheumatoid”
Line 401 - Table 1: please also categorize the seaweeds per color (red, green and brown) as discussed in text and done in table 2 for the cultivation methods (Inland and Ocean-based cultivations).
Line 408 - Please replace “Cauleurpa” by “Caulerpa”
6.Genetic Manipulation and Strain Improvement
Line 527 - Please replace “euchematoid” by “eucheumatoid”
8.Prospects and Recommendations
Line 785 - Please replace “lactuva” by “lactuca”
Best Regards,
Reviewer
Reviewer 3 Report
The manuscript by Rajeena Sugumaran et al reports “A Retrospective Review of Global Commercial Seaweed Production – Current Challenges, Measures and Prospects”. This review discusses biosecurity methods addressing seaweed industry difficulties which are pushing increasing quantity and quality of algal biomass to meet existing challenges, as well as talks about prospects for seaweed research. Latest biosecurity developments from several segments in the seaweed research (234 references) are well presented in this review. Overall, this review is quite large (Almost 20,000 words with 9 sections), but it discusses well the topic that the authors have mentioned in detail. The manuscript was written in a very careful way. I highly recommend publishing this interesting review as it captures the attention of the readers of the International Journal of Environmental Research and Public Health.
Some concerns and suggestions are listed below for the author's attention.
Comment 1: Please improve Figures 1 and 3 if possible by increasing the text size to be more readable.
Comment 2: Figure 2 will be more interesting if you provide a zoomed image, in order to show more details. if it is possible.
Comment 3: The English language was written in a very careful way however I found some minor typos. You can find the grammatically corrected manuscript in the attached file.

Reviewer 4 Report
The manuscript entitled "A Retrospective Review of Global Commercial Seaweed Production – Current Challenges, Measures and Prospects" addresses a relevant and appropriate topic for this journal.
A thorough taxonomic review must be introduced before being accepted for publication.
Corrections needed:
line 38/49 - which includes red seaweeds (Eucheuma spp., Kappaphycus alvarezii, Gracilaria spp., Porphyra spp.); brown seaweeds (Saccharina japonica, Undaria pinnatifida, Sargassum fusiforme); and green seaweed (Ulva clathrata, Monostroma nitidum, Caulerpa spp.)
line 62/64 - ... there was an increase in emissions of greenhouse gases, such as CO2 and enteric methane (CH4), by 18 and 21%, respectively, while ocean temperatures rose by 0.13°C ...
line 118 - ... by Kappaphycus alvarezii sap provides ...
line 135 - seaweed Ulva sp. can be a feedstock ...
line 239 - ... Saccharina religiosa (formerly Laminaria religiosa), in Japan ...
line 240 - ... Fucus distichus subsp. evanescens (Fucus evanescens), ...
line 267 - ... Polysiphonia and Melanothamnus (formerly Neosiphonia) ...
line 278 - found, Melanothamnus apiculatus (formerly Neosiphonia apiculata) (Semporna) and Melanothamnus savatieri (formerly Neosiphonia savatieri (Kudat) (Rhodophyta), with high infection rates during
line 283/285 - ... cultured Kappaphycus seaweed with free-floating ‘EFA infected’ Sargassum sp., and second, through imported cultivation stocks that were infected ... (Note: "Kappaphycus" in italics, please)
line 286 - ... and Laurencia dendroidea (formerly Laurencia majuscula), have ...
line 290 - Figure 5. Signs of Neosiphonia infection on K. alvarezii branch harvested from Kota Belud, Sabah. (Note: "Neosiphonia" in italics, please)
line 306 - ... and Sargassum spp. (Note: "spp." is not in italics)
line 345/346 - ... on Sargassum filipendula [102], Peraphithoe parmerong on Sargassum linearifolium [103] and Cymodocea japonica on Undaria pinnatifida (Phaeophyceae) has appeared ...
line 314 - infections from the family Phycodnaviridae have been ... (Note: "Phycodnaviridae" is not in italics)
line 323 - ... in Korean Pyropia farms, ... (Note: "Pyropia" is in italics)
line 325 - ... a study in Japanese Pyropia (Note: "Pyropia" is in italics)
line 345 - ... like Ampithoe longimana on Sargassum filipendula ... (Note: authors should review this part of the manuscript, as it appears to be a repetition of previous paragraphs)
line 404 - ... Green seaweeds of the genus Caulerpa
line 408 - ... Caulerpa lentillifera and Caulerpa racemosa, which ...
pg. 13, Table 2 - Gracilariopsis longissima (formerly Gracilaria verrucosa)
pg. 13, Table 2 - Gracilariopsis lemaneiformis (Gracilaria lemaneiformis)
line 476 - ... and Gracilariopsis lemaneiformis (Gracilaria lemaneiformis) in ...
line 577 - .... Saccharina longissima (formerly Laminaria longissima) and S. japonica, ...
line 705 - ... Lyngbya majuscula (Cyanobacteria), which ...
line 785 - ... cultivated green algae, Ulva lactuca, was ...
line 786 - ... The red algae, Gracilaria dura, showed ...
line 794/795 - ... K. alvarezii revealed no ... (Note: "revealed" is not in italics)
